# Social media strategies used to translate knowledge and disseminate clinical neuroscience information to healthcare users: A systematic review

**Rachel Victoria Baran**[1], **Melissa Fazari**[1], **David Lightfoot**[2], **Michael David Cusimano**[1]*

**1** Injury Prevention Research Office, Division of Neurosurgery, St. Michael's Hospital, Toronto, Ontario, Canada, **2** Health Sciences Library, St. Michael's Hospital, Toronto, Ontario, Canada

* injury.prevention@unityhealth.to

## Abstract

Social media can be an important source of clinical neuroscience information for healthcare users (e.g., patients, healthcare providers, the general public). This systematic review synthesized evidence on the effectiveness of social media strategies in translating knowledge and disseminating clinical neuroscience information to healthcare users. A systematic review of six electronic databases up to July 29, 2024 was conducted. Original, peer-reviewed articles examining the effectiveness of YouTube, Facebook, LinkedIn, Twitter, social media messaging apps, or a combination of these platforms in translating clinical neuroscience information to healthcare users (e.g., patients, healthcare providers, caregivers, and the general public) were eligible for inclusion. Several proxies (e.g., change in uptake of research, change in awareness, change in knowledge, change in understanding, behaviour change, and/or change in social media metrics) were considered as outcomes of knowledge translation (KT) effectiveness. Two independent reviewers screened articles and assessed risk of bias. The protocol was registered on PROSPERO (ID: CRD42021269034). A total of six studies were included in this review. The included studies used YouTube, Facebook, Twitter, or a combination of social media platforms aimed at healthcare users. Most social media strategies used to disseminate clinical neuroscience information in the included studies (N = 5/6) resulted in improved indicators of KT. However, due to the high risk of bias among the included studies, these results must be interpreted with caution. Disseminating clinical neuroscience information via Facebook, Twitter, YouTube, or a combination of these platforms may achieve the goals of KT. However, there is currently a gap in the literature about clinical neuroscience KT via social media, both in the quantity of studies and quality of evidence. Future research should aim to minimize the risk of bias by controlling for important confounding factors and use objective measures of KT to complement subjective measures.

## Author summary

Social media is increasingly being used to communicate health information, and specifically clinical neuroscience information (e.g., concussion, stroke) to the general public,

**Data availability statement:** All data relevant to this review are included in the article. Any requests for the template data collection forms or risk of bias forms should be made by correspondence with the corresponding author via email (injury.prevention@unityhealth.to).

**Funding:** The author(s) received no specific funding for this work.;

**Competing interests:** The authors have declared that no competing interests exist.

patients, and healthcare providers. However, the social media strategies that are best suited to translate clinical neuroscience information are currently unknown. Understanding evidence-based knowledge translation strategies employed via social media are critical so that knowledge producers (e.g., researchers, clinicians) can appropriately convey clinical neuroscience information in a way that is understood and used by their target audience. Here, we conducted a systematic review of articles examining the effectiveness of social media in the knowledge translation of clinical neuroscience information. We included a total of six studies in this review. We found that strategies disseminating information via Facebook, Twitter, YouTube, and a combination of these platforms may be effective at the knowledge translation of clinical neuroscience information. However, the low quality of evidence in the few included studies prevents any firm conclusions about the effectiveness of clinical neuroscience knowledge translation strategies via social media. We also found a need for future research on the effectiveness of various knowledge translation strategies in disseminating clinical neuroscience content via social media. Our review provides new insights into evidence-based knowledge translation strategies via social media that researchers and clinicians may use to effectively disseminate clinical neuroscience information and reach their intended audience.

## Background

Social media platforms (e.g., Facebook, Twitter (also known as X), YouTube, Instagram) have the capacity to reach many individuals and rapidly communicate information. Many researchers and patient organizations often conduct knowledge translation (KT) activities via social media to provide end users such as patients, healthcare providers (HCP), and the general public with reliable sources of health information. In recent years, the number of social media users has been increasing and with the COVID-19 pandemic presenting barriers to in-person health information dissemination, social media has been increasingly used to communicate health information among researchers, HCP, and the general public [1,2].

KT is defined by Health Canada as "a dynamic and iterative process that includes synthesis, dissemination, exchange, and ethically-sound application of knowledge" occurring "between researchers and knowledge users" [3]. KT encompasses multiple outcomes such as the extent to which information is distributed and engaged with (e.g., media views, comments), the usefulness of the KT initiative (e.g., gained awareness or knowledge), and whether the knowledge is used (e.g., behaviour or policy change) [4]. While social media has been increasingly used for translating knowledge, especially among researchers [5,6], the effectiveness of using social media to translate knowledge about a specific topic in healthcare has not been reviewed in the literature [6]. Social media presents unique challenges to KT, as users face a surplus of health information which may be difficult to navigate or understand. Given this, it is imperative to identify effective KT strategies for disseminating information to target audiences on social media platforms, which may be especially useful for researchers.

Currently, the body of literature on clinical neuroscience (e.g., traumatic brain injury (TBI), concussion, dementia, brain tumours, etc.) is large and growing at a rapid pace. Several studies have shown that social media is a source of information for many conditions relating to clinical neuroscience [7,8]. Studies have identified Twitter as an important source of sports concussion information and education, and one of the main media sources for high attention papers on Parkinson's Disease [8,9]. In addition, patients with neurological conditions use social media platforms to share information and raise awareness about their condition [7,10,11]. For example, one study found that patients with TBI used Twitter to communicate

and raise awareness about TBI and sport concussions [7]. Furthermore, there is a demand for social media users to receive information on social media about topics in the field of clinical neuroscience [12]. This demonstrates an opportunity for knowledge producers to tap into these social media networks and disseminate clinical neuroscience information using social media. Evaluating the effectiveness of KT strategies via social media is especially important in the field of clinical neuroscience, as many of these patient populations suffer from altered cognition, which may impair their ability to use social media and impose additional limitations to successful KT [13]. Therefore, it is imperative to identify evidence-based social media KT strategies, so that knowledge producers in the clinical neuroscience field can be equipped with the tools to effectively disseminate information and achieve the goals of KT.

Although multiple reviews have explored the effectiveness of KT strategies, including a review about KT strategies specific to a topic in the clinical neurosciences (e.g., concussions), no review has evaluated the effectiveness of clinical neuroscience KT initiatives disseminated via social media [14,15]. This systematic review aims to fill this gap by synthesizing the research that currently exists regarding social media strategies that aim to translate clinical neuroscience information to knowledge users.

Our review aimed to answer the following questions:

1. Which social media strategies have been used to translate knowledge and disseminate clinical neuroscience information to healthcare users (e.g., patients, HCP, and the general public)?

2. How effective are these strategies at achieving knowledge translation of clinical neuroscience information to healthcare users?

This review will provide a summary and evaluation of peer-reviewed primary research studies in medical databases that assess current social media strategies used to translate clinical neuroscience information to healthcare users. Based on the results of this literature review, we intend to better understand clinical neuroscience KT activities disseminated via social media, assess their effectiveness, and identify potential gaps and areas for future research within this field.

## Methods

### Protocol, eligibility criteria and ethics approval

The search strategy and protocol used to select studies eligible for inclusion were developed using the PICO (Participants, Interventions, Comparators, and Outcomes) framework. Participants of interest were healthcare users, which, for the purposes of this review, were defined as patients, HCP (e.g., physicians), caregivers, and the general public. The intervention of interest was any KT intervention that used one or more of the following social media platform(s): Twitter, Facebook, YouTube, LinkedIn, and/or social media messaging apps (e.g., WhatsApp) to disseminate clinical neuroscience information, including, but not limited to TBI, concussion, dementia, Alzheimer's disease, multiple sclerosis (MS), and brain tumours. Any studies that used Short Message Service (SMS) and/or text messaging were excluded, as these were not deemed types of social media platforms. No restrictions were placed on the type of comparator eligible for inclusion. Examples of comparators that were considered relevant for inclusion were the absence of a KT intervention or a KT intervention that did not use social media. The outcome of interest was the effectiveness of the social media strategy in translating clinical neuroscience knowledge to healthcare users. For the purposes of this review, multiple indicators, which were derived from the Alberta Health Services Knowledge Translation Evaluation Planning Guide, were considered proxies to measure the impact of KT activities [4].

The proxies for KT effectiveness that were eligible for inclusion were the change in uptake of research or information, change in awareness, change in knowledge, change in understanding, behaviour change, and/or change in social media metrics (e.g., number of views, likes, shares, etc.). While many indicators of KT are subjective perceptions of knowledge gains, the authors included participant self-reported indicators of KT in the review while acknowledging their limitations in the analysis of results, so that the full scope of research in this field would be captured in this review. No restrictions were placed on the study setting or design eligible for inclusion. All included studies were original, peer-reviewed, reported in the English language, and had a publication year of 2004 or later as the social media platforms included in this review became available in or after 2004.

The systematic review protocol was registered with the International Prospective Register of Systematic Reviews (PROSPERO) in Fall 2021 (Registration ID: CRD42021269034). An amendment to the protocol was made to broaden the outcomes of interest to include change in social media metrics (e.g., number of views, likes, shares) on posts with information or research about clinical neuroscience. This outcome was added as social media metrics were deemed a valid measurement of the effectiveness of KT strategies after further consideration and based on reports in the literature. The updated protocol, including this additional outcome of interest, was resubmitted on PROSPERO.

No ethics approval was required for this systematic review.

## Literature search and selection

An Information Specialist at St. Michael's Hospital (DL) co-developed the search strategy and conducted the original search in the following online databases: Ovid MEDLINE, Embase, CINAHL, Cochrane Database of Systematic Reviews, Scopus, and Web of Science on July 16, 2021. These medical databases were selected because this review focused on the dissemination of information and/or research specifically pertaining to the clinical neurosciences. The full search strategy is outlined in S1 File. Bi-weekly email updates from the Ovid MEDLINE search strategy were received until July 29, 2024, to ensure that the review was up to date.

The search results from the electronic databases and the bi-weekly updates were imported into Covidence, a web-based software used to manage and screen all abstracts and full-text articles. The abstract screen was completed by two independent reviewers (RVB and MF). To reduce errors in screening, both reviewers were trained prior to the selection process on the PICO framework and inclusion/exclusion criteria. A Cohen's kappa of 1.0, which indicated perfect agreement between the two reviewers, was achieved before proceeding to the full-text screen to ensure inter-rater reliability.

The same two independent reviewers (RVB and MF) conducted a full-text screen of all relevant abstracts to assess each study's eligibility for inclusion in the review based on the PICO criteria. A third reviewer was not required as there were no discrepancies at the abstract or full-text screening stage.

## Quality of evidence

Two independent reviewers (RVB and MF) performed a risk of bias (ROB) assessment for all relevant outcomes in each included study. Two different ROB assessment tools were used to assess the risk of bias since the appropriateness of the tool depends on the study design: (1) Cochrane Risk of Bias in Non-randomised Studies – of Interventions (ROBINS-I) tool was used to assess the ROB in outcomes for included non-randomized studies [16] and (2) Revised Cochrane risk of bias tool for randomized trials (RoB2) was used to assess risk of bias in the one randomized controlled trial included [17]. For ROBINS-I, an overall ROB judgement of

"low", "moderate", "serious", or "critical" was assigned to each outcome of interest based on the *seven* domain-level judgements regarding ROB: (1) confounding, (2) selection of participants, (3) classification of interventions, (4) deviations from intended interventions, (5) missing data, (6) measurement of outcomes, and (7) selection of the reported result [18]. For RoB2, an overall ROB judgement of "low", "high", or "some concerns" was assigned to the outcome of interest based on the *five* domain-level judgements: (1) randomization process, (2) deviations from intended interventions, (3) missing outcome data, (4) measurement of the outcome, and (5) selection of the reported result [19]. Both reviewers independently reviewed each included study outcome across all bias domains according to the full-text article. Any reviewer discrepancies in domain judgements were resolved through discussion and referring to the methodology described in the full-text article, which resulted in consensus being achieved. There were no discrepancies between reviewers regarding the overall ROB judgements for each outcome.

## Data collection and narrative synthesis

The following data were extracted from each included study by two independent reviewers (RVB and MF) and stored in Microsoft Excel: study author(s), year of publication, study title, study design, study location, data sources/years, PICO items, statistical methods, and relevant results. A narrative synthesis was performed to systematically summarize the extracted data from all included studies. The summarization of data for the narrative synthesis was done manually and collectively by two reviewers (RVB and MF) based on the full-text and ROB assessments for each outcome.

## Results

### Screening results

The initial database search yielded 4543 relevant studies. Bi-weekly email updates from Ovid MEDLINE up to July 29, 2024, yielded another 752 studies. After the removal of duplicates (N = 1350), 3945 abstracts were identified and screened. Of 3945 abstracts screened, 23 studies were deemed relevant to undergo a full-text screen to be assessed for eligibility. After the full-text screen, 17 articles were excluded. Some reasons for exclusion were that these studies conducted a content analysis of clinical neuroscience information available on social media platforms rather than evaluating their effectiveness on KT [20,21], and interventions disseminated content via both social media and other non-social media methods (S1 Table and Fig 1). The study by Narayanaswami et al. (2015) that compared traditional, non-social media methods of dissemination alone to the addition of social media methods of dissemination was included. This study was deemed relevant for inclusion in our review because this comparison examines how the addition of social media KT methods impacts KT indicators, whereas other excluded studies did not compare this to the impact of non-social media methods alone [22]. Among the six included studies, 15 outcomes were evaluated for risk of bias as some studies explored multiple outcomes.

### Risk of bias assessment

The ROBINS-I tool was used to assess the ROB of the outcomes in three non-randomized uncontrolled before after studies and in two non-randomized studies of intervention [16]. All outcomes of interest (N = 14) from the non-randomized studies were deemed as having an overall judgment of "serious" ROB because at least one domain (the one pertaining to confounding) was judged as having "serious" ROB based on Cochrane Guidelines (S1 and S3 Figs) [18]. This was because no study appropriately controlled for many of the important

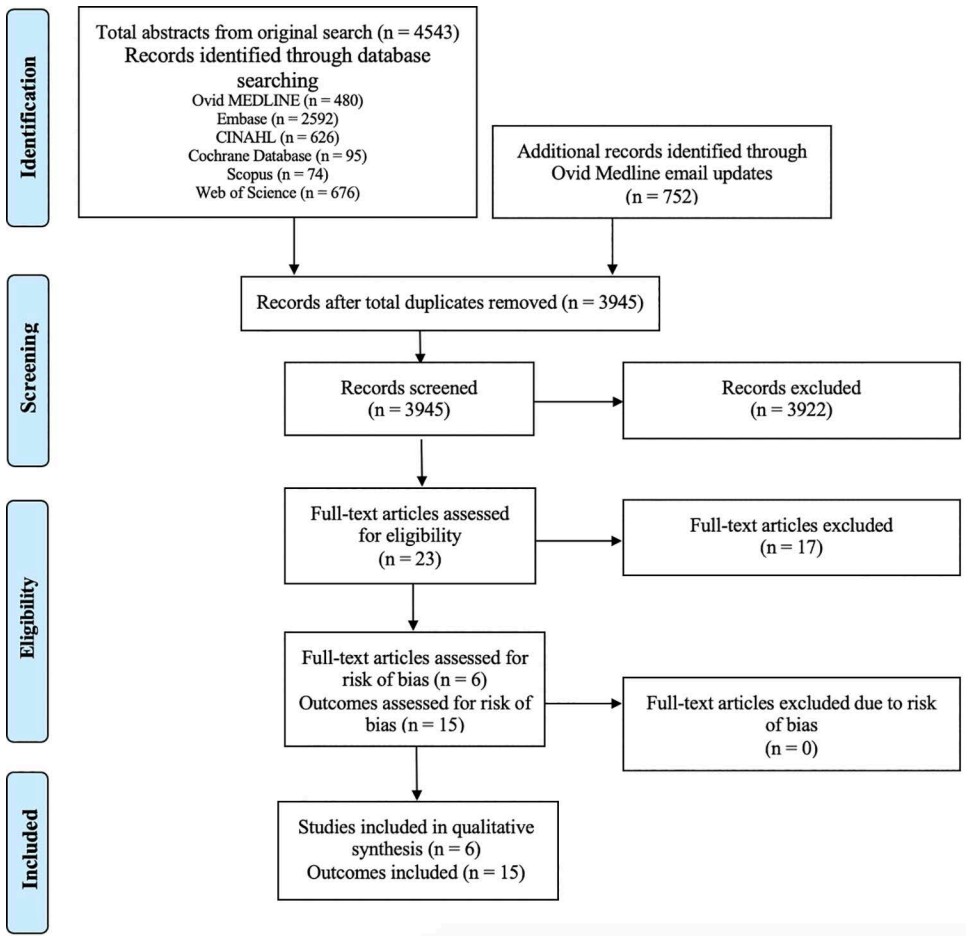

**Fig 1. CONSORT flow diagram of study selection for systematic review based on the Preferred Reporting Items for Systematic Reviews and Meta-Analyses (PRISMA) Flow Diagram for systematic reviews.** Figure modified from the PRISMA CONSORT diagram template to reflect full-text article eligibility based on risk of bias assessment. See S1 Table for reasons for excluding articles at the full-text screening stage.

factors (e.g., age, sex, baseline user familiarity with social media platforms, etc.), which could confound the effect of the intervention on the outcomes.

For the individually randomized controlled trial (parallel-group design), there was only one outcome of interest, which was deemed to have "some concerns" overall according to the RoB2 tool (S2 and S4 Figs) [17]. There were concerns with regards to bias arising from the randomization process and bias due to deviations from intended interventions, as participants and researchers were unblinded [23].

## Characteristics of included studies

Three of six included studies used a non-randomized uncontrolled before after study design, two were non-randomized studies of intervention, and one was a randomized controlled trial (Table 1). Five of the six studies were quantitative in nature, and one study gathered qualitative data via interviews with participants. The included studies assessed different outcomes as proxies to measure the effectiveness of the intervention (S5 Fig). Change in knowledge was the most frequently assessed outcome (N = 4/15), followed by change in awareness (N = 3/15), behaviour change (N = 3/15), and change in social media metrics (N = 3/15). Improved

**Table 1. Characteristics of the six studies included in the systematic review.**

| Authors | Study Design | Population | Participants | Intervention | Comparator | Outcome | General Findings |
|---|---|---|---|---|---|---|---|
| Ahmed et al 2017 [24] | NR uncontrolled before after | Patients with a concussion | N = 11 9 M, 2 F Mean age = 21 | 3-month Facebook concussion education program | Pre-intervention | O1: # of patients with increased self-reported concussion knowledge O2: # of patients with self-reported behaviour change for RTP decision | N = 10/11 (91%) N = 9/11 (82%) |
| Castillo et al 2021 [25] | NR uncontrolled before after | O1: General public O2: General public O3: Caregivers of patients O4: Caregivers, HCP, public | N = 34 88.6% F, 11.2% M Mean age = 48.6 | 5-month Twitter campaign about PiD YT video about PiD posted for 5 months YT video about PiD YT video about PiD | 5-month pre-campaign period Pre-YT video Pre-YT video Pre-YT video | O1: Mean # of tweets per month about PiD during campaign vs pre-campaign O2: Mean degree of awareness about PiD post-video vs pre-video O3: Mean improved understanding about PiD post-video O4: Mean level of new information acquired post-video | Campaign: 265.20 (SD = 82.27) vs pre-campaign: 141.40 (SD = 41.60) p = 0.017 Post-video: 6.15 (SD = 1.19) vs pre-video: 3.12 (SD = 2.10) 5.68 (SD = 1.89) Caregiver: 5.53 (SD = 1.93) HCP: 5.78 (SD = 1.38) Public: 5.84 (SD = 1.49) |
| Castillo et al 2024 [26] | NR study of intervention | Knowledge users (patients, caregivers, HCP, policymakers) | N = 13 O4: 476 F, 54 M Mean age = 67.39 | 12-month campaign about pain in dementia via Twitter, Instagram, Facebook, and LinkedIn | None | O1: # of social media metrics O2: Change in awareness about PiD O3: Change in knowledge about PiD O4: Mean rating of behaviour change (7-point scale) | O1: see section 'Change in social media metrics' O2: see section 'Change in awareness' O3: see section 'Change in knowledge' O4: see section 'Behaviour change' and below Patient: 5.10; Caregiver: 4.90; HCP: 4.84; Public: 4.33 |
| Chan & Leung 2020 [23] | RCT | HCP | N = 80 61 F, 19 M Age range = 20-69 | 8-week Facebook dementia education program | 8-week email dementia education program | Mean change in dementia knowledge score (DKAS) | Communication & Behaviour subscale of DKAS: Facebook = 2.1 vs email = 0.3, 95% CI [0.4, 3.3] (p = 0.02) Total DKAS scores, other DKAS subscale scores, multiple choice questions: no statistically significant differences (p > 0.05) |
| Narayanaswami et al 2015 [22] | NR uncontrolled before after | Patients with MS, physicians of patients with MS | N = 800 patients (mean age = 54.8) N = 2480 physicians (mean age = 52.1) | 1-month dissemination of CPG content on CAM in MS via Facebook, Twitter, YT, LinkedIn, Neurology audio podcast | 1-month dissemination of CPG content on CAM in MS via print, email, and the Internet | O1: AD in knowledge changes about CAM in MS O2: AD in behaviour changes regarding CAM in MS O3: AD in awareness changes about CAM in MS | Patients: -4%, 95% CI (-14, 6) Physicians: -6%, 95% CI (-16, 5) Patients: 1%, 95% CI (-14, 15) Physicians: -6.4%, 95% CI (-16, 3) Patients: -4%, 95% CI (-6, 14) Physicians: -7%, 95% CI (-18, 4) |
| Zheng & Woo 2017 [27] | NR study of intervention | General public | N/A | 2 YT videos about dementia posted for 17 months | 5 talk-based workshops about dementia | Number of YT video views vs number of workshop attendees | YT views: 2505 vs workshop attendees: 744 |

**Notes:** AD=absolute difference; CAM=complementary alternative medicine; CI=confidence interval; CPG=Clinical Practice Guidelines; DKAS=Dementia Knowledge Assessment Scale; F = females; HCP=healthcare providers; M = males; MS=Multiple Sclerosis; NR = non-randomized; O1, O2, O3, etc…=outcome 1, outcome 2, outcome 3, etc…; p=p-value; PiD = Pain in Dementia; RCT=Randomized Controlled Trial; ROB=risk of bias; RTP=Return-to-Play; SD=standard deviation; YT = YouTube

understanding and uptake of research were the least frequently assessed outcomes among the included studies with only one study assessing each outcome. Among the included studies, outcomes were assessed among patients [24], the general public [25,27], HCPs [23], caregivers

of patients [25], and combinations of multiple population groups [2226]. For example, Castillo et al. (2021) assessed the mean level of new information acquired post-video among caregivers, HCP, *and* the general public and Narayanaswami et al. (2015) assessed each of the three outcomes among patients with MS and physicians of patients with MS (Table 1) [22,25].

Among all included outcomes, a combination of multiple social media platforms were most frequently used to disseminate clinical neuroscience information (N = 7/15) (S6 Fig). Two studies examined the effectiveness of a combination of Facebook, Twitter, YouTube, Instagram, and/or LinkedIn amongst others on change in knowledge, change in awareness, and behaviour change [22,26]. Other studies used YouTube (N = 4/15), Facebook (N = 3/15), or Twitter (N = 1/15) alone to disseminate clinical neuroscience information. The majority of interventions disseminated information on dementia (N = 10/15), while the others disseminated information on MS (N = 3/15) and concussion (N = 2/15) (S7 Fig). All the included studies implemented their interventions for different durations of time, ranging from one to 17 months.

## Effectiveness of the intervention

Overall, most studies (N = 5/6) reported that the social media strategies used to disseminate clinical neuroscience knowledge to the target population were effective at achieving the goals of KT. Only one study, which used multiple social media platforms in addition to traditional, non-social media methods to disseminate Clinical Practice Guidelines (CPG) on complementary alternative medicine in MS over one month, was not effective in improving KT indicators compared to disseminating this content via traditional methods alone [22]. However, due to the insufficient number and lack of methodological rigour of the included studies, as demonstrated by the serious risk of bias in all non-randomized studies, conclusions about the overall effectiveness of clinical neuroscience KT strategies via social media cannot be determined.

## Change in knowledge

Increased knowledge was the most common outcome measured (N = 4/15) (S5 Fig). This outcome was evaluated by four studies and was assessed by either a self-reported measure of knowledge [24], survey questions about the disseminated content [22], a semi-structured interview [26], or a valid and reliable measure of dementia knowledge, the Dementia Knowledge Assessment Scale (DKAS) [23]. Ahmed et al. found a three-month Facebook concussion education program to be effective at increasing self-reported concussion knowledge among patients with a concussion, as demonstrated by N = 10/11 patients having reported that their knowledge of concussion increased [24]. An eight-week Facebook education program aimed at HCPs compared to an email education program increased HCPs knowledge on dementia among all DKAS subscales, but only had a statistically significant difference (p = 0.02) in the Communication and Behaviour DKAS subscale [23]. Castillo et al. (2024) found that their 12-month campaign via multiple social media platforms about pain in dementia positively impacted knowledge gains among participants. Healthcare providers and caregivers felt that they could apply this knowledge to their clinical practice and when assessing pain for family members with dementia respectively [26]. Narayanaswami et al. (2015) disseminated CPG content over one month via multiple social media platforms in addition to traditional methods of dissemination (e.g., print, email, the Internet). However, the addition of social media methods was not shown to be effective at increasing knowledge among patients with MS and physicians of patients with MS compared to traditional methods alone (Table 1) [22].

## Change in awareness

Three studies measured change in awareness via survey questions or a semi-structured interview [22,25,26] (S5 Fig). Castillo et al. (2021) found an increase in the general public's self-reported awareness about pain in dementia, as measured by a Likert scale, after watching a YouTube video [25]. These authors subsequently expanded the dissemination of pain in dementia information to a 12-month multi-social media campaign and assessed self-reported changes in awareness among knowledge users (e.g., patients, HCPs, caregivers) [26]. The majority of participants reported that the campaign increased awareness about pain in dementia, with some participants attributing this to the way that the information was conveyed (e.g., a short, animated video of a person with dementia being aggressive when in pain). Another study compared one month of disseminating CPG information about complementary alternative medicine in MS via multiple social media platforms in addition to print, email, and the Internet versus dissemination via print, email, and the Internet alone [22]. This additional method of disseminating content via social media was ineffective at increasing awareness among patients with MS (-4%, 95% CI [-6, 14]) and physicians of patients with MS (-7%, 95% CI [-18, 4]) compared to traditional dissemination methods alone (Table 1).

## Change in social media metrics

Three of the six studies evaluated change in social media metrics [25–27]. Each of the three studies disseminated information about dementia, with two of the studies focusing on the general public [25,27] and the other on knowledge users (e.g., patients, HCPs, caregivers) [26]. One study used Twitter to assess the mean number of tweets posted about pain in dementia during the five-month social media campaign compared to a five-month pre-campaign period [25]. The Twitter social media campaign was effective at increasing social media metrics as demonstrated by the increased number of tweets during the campaign (265.20) compared to the pre-campaign period (141.40), which was a statistically significant difference (p = 0.017). Castillo et al. (2024) subsequently expanded the pain in dementia campaign over 12 months and included multiple platforms (e.g., Twitter, Facebook, Instagram, LinkedIn, videos on YouTube). While they did not compare metrics before versus during or after the campaign, they reported that the Facebook page reached 1, 313, 485 users, the short, animated videos received over 257,000 views, and the pain in dementia hashtag was used in 2835 posts, reaching over 1.5 million users. Another study used two YouTube videos on dementia to assess video views compared to the number of attendees at five in-person workshops presenting the same content on dementia [27]. The YouTube videos were similarly effective at increasing social media metrics as they received 2505 views compared to the 744 attendees at the in-person workshop, but it is unknown if this difference is statistically significant.

## Behaviour change

Behaviour change was assessed in three studies and measured by participants' self-reported behaviour change (S5 Fig). One study found that most patients with a concussion (N = 9/11) reported that a three-month Facebook concussion education program assisted in their decision to return to sport after sustaining a concussion [24]. The study by Castillo et al. (2024) found that the social media campaign on pain in dementia influenced HCPs advocacy and provision of care in their clinical practice. Additionally, the campaign influenced participants self-reported information-seeking behaviour as rated on a seven-point scale, with people living with dementia being most likely to seek additional information (5.10), then caregivers (4.90), then HCPs (4.84), followed by the public (4.33). However, another study found that after disseminating CPG via multiple social media platforms and traditional methods,

compared to traditional methods alone, patients and physicians were not more likely to discuss this content in a clinical setting (Table 1) [22].

## Uptake of research and change in understanding

One study implemented a YouTube video on pain in dementia and measured the change in uptake of research and change in understanding via a questionnaire with Likert scale questions (scale ranging from 1 to 7). The video was found to be effective in improving mean understanding about pain in dementia among caregivers of patients, and similarly effective at improving the mean level of new information acquired among caregivers, HCPs, and the general public (Table 1) [25].

## Discussion

This systematic review examined various social media strategies that have been used to translate clinical neuroscience knowledge to healthcare users (e.g., patients, caregivers, HCPs, and the general public). The effectiveness of these social media KT strategies as measured by indicators of KT (e.g., change in awareness, knowledge, understanding, social media metrics, uptake of research and/or behaviour change) was also reviewed among the included studies. While five of the six included studies found that the social media strategies used were effective in translating knowledge about a topic in the clinical neurosciences to the target population, the overall results of this review remain inconclusive. First, there were an insufficient number of studies available in the literature and eligible for inclusion in this review to be able to draw accurate conclusions on the effectiveness of the social media KT strategies used. Second, there was a lack of methodological rigour among the included studies, as demonstrated by the serious risk of bias judgement designated to all non-randomized studies (N = 5/6), which precludes any definitive conclusions about the effectiveness of certain social media strategies in translating clinical neuroscience information. Third, the measurement of KT indicators in the included studies were often subjective perceptions of indicators such as gained awareness or knowledge, making it difficult to capture and report on the true effectiveness of the interventions on KT. Therefore, due to the small number of included studies, poor quality of evidence, and subjective measures of KT in the included studies, any conclusions on the effectiveness of clinical neuroscience KT strategies via social media must be interpreted with caution.

Similar to previous studies which examined Twitter and Facebook platforms [7–9], our review found that these two platforms can serve as important sources of clinical neuroscience information for patients, HCPs, and the general public [23–25]. Additionally, both Twitter and YouTube were found to be effective at increasing social media metrics [25,27]. Our review suggests that disseminating clinical neuroscience information via Facebook, Twitter, YouTube, or a combination of multiple social media platforms (e.g., Twitter, Instagram, Facebook, and LinkedIn) for a minimum of two months may be effective at translating knowledge to patients, HCPs, caregivers, and the general public.

Furthermore, based on our review, the studies which demonstrated improved indicators of KT posted clinical neuroscience content frequently and consistently [23–25], and disseminated content that was appealing and interactive [23–25,27]. For example, some of the interactive strategies included the use of images, educational videos, diagrams, online games, hashtags, and/or provided users with the opportunity to obtain feedback from medical professionals [23–25,27]. Therefore, researchers may be able to leverage both interactive content and behaviour change techniques (e.g., two-way communication, the use of a credible source) to effectively translate clinical neuroscience knowledge to the intended audience [23,24,28].

Only one study found no change in awareness, knowledge, and behaviour after the dissemination of CPG content via multiple social media platforms in addition to traditional methods (e.g., email, print, the Internet), compared to disseminating this content via traditional methods alone among patients with MS and physicians [22]. However, the additional dissemination of CPG content via social media occurred three months after this content was published and disseminated via print, email, and the Internet. Therefore, physicians and patients may have been previously exposed to the content, which may explain why there were no increases in awareness, knowledge or behaviour change with the addition of social media dissemination [22]. While this study by Narayanaswami et al. (2015) examined the addition of social media methods of dissemination compared to traditional methods alone, all other included study interventions used social media dissemination methods alone. Therefore, due to the inherent differences in study designs and comparators, the effectiveness of social media KT strategies for the clinical neurosciences cannot be reasonably compared between the study by Narayanaswami et al. (2015) and the other included studies.

It is also plausible that the lack of changes seen across outcomes in the study by Narayanaswami et al. (2015) may be explained by the short time period of dissemination (one month) compared to the other studies included. This is consistent with findings from Elliot et al., who reported that the duration of social media KT strategies and consistent posting should be considered because building online communities to receive content can be a difficult and lengthy process [29]. Second, the authors of the study by Narayanaswami et al. (2015) did not tailor the disseminated content based on the type of social media platform used and instead disseminated the same content via multiple social media platforms [22]. This strategy may have been ineffective because each social media platform has a different purpose and unique functionality [29]. For example, Facebook may be best used to facilitate discussions with patients to improve their knowledge, whereas Twitter may be more effective at broad dissemination of information to improve awareness among the general population. Tailoring content to the social media platform was demonstrated effectively by Ahmed et al. (2017) and Chan and Leung (2020), who used an interactive approach and facilitated discussions to disseminate content via Facebook [23,24].

It is important to note that many of the included studies (N = 5/6) did not control for important confounding factors, including baseline user familiarity with social media platforms, age, and sex, and thus were deemed to have "serious" risk of bias. User familiarity with social media platforms should be controlled for because individuals with greater experience using social media may access and understand information presented on the platform with greater ease compared to novel social media users. For the one randomized controlled trial, the study outcome of mean change in knowledge was deemed as having "some concerns" overall due to potential bias from the randomization process and deviations from the intended intervention. This is because it was unknown whether the allocation sequence was concealed until participants were assigned to their intervention, and participants and individuals delivering the intervention were unblinded. Future studies should control for important confounding factors such as these to minimize risk of bias.

## Limitations and strengths

Our review has some limitations which affect the interpretation and generalizability of our results. First, only a small number of articles were included due to the overall lack of published work on the KT of clinical neuroscience information via social media and stringent inclusion criteria. This is because many research papers examined SMS strategies to disseminate clinical neuroscience content or examined the clinical neuroscience content available on social media

platforms and thus were not eligible for inclusion in this review. Additionally, while this review focused the literature search on medical science databases, potential studies relevant for inclusion in social science databases may have been missed. Despite the few numbers of included studies, multiple outcomes were reported in many of the six included studies, which allowed for the comparison and evaluation of 15 study outcomes.

Second, some of the included studies had small sample sizes (<15 participants) [24], which makes it difficult to determine whether the results observed were due to chance alone. Another limitation is that multiple proxy measures such as the uptake of research, awareness change, social media metrics, knowledge change, and behaviour change were used as indicators of KT. The lack of one standard measurement of KT effectiveness limits the comparability of results. Furthermore, several outcomes used self-reported data to assess knowledge and behaviour change rather than objective methods of measuring these outcomes. Self-reported data is susceptible to response bias and social desirability bias and may not be representative of actual knowledge and behaviour change. A final limitation is the use of social media metrics as an indicator of KT. For example, one study relied solely on the number of YouTube views, which may not be indicative of whether KT was effective since views may include users who only watch a portion of the video and singular users who watch the same video multiple times [27,30,31]. Despite these limitations, our review has strong methodological rigor and high inter-rater reliability between the two reviewers.

## Future studies

To our knowledge, this is the only review to investigate the effectiveness of translating clinical neuroscience knowledge via social media. However, due to the small number of included studies and the lack of control for confounding among these studies, future research that considers these factors should be conducted to elucidate our results. This review revealed that there is a paucity of peer-reviewed literature about clinical neuroscience KT strategies disseminated via social media, that is methodologically rigorous and provide high-quality evidence. Therefore, future studies should include outcomes which directly measure KT instead of only relying on social media metrics and subjective measures such as awareness change, which do not imply that knowledge has been gained and/or applied to practical use. Furthermore, objective measures of knowledge and/or behaviour change should complement self-reported measures. Future research should expand on the results of our review by comparing social media dissemination interventions to other non-social media dissemination interventions, comparing different social media interventions to each other, and/or comparing combinations of different social media dissemination interventions and/or non-social media dissemination interventions. Additionally, while this systematic review focused on peer-reviewed, original research articles, future studies should consider expanding into grey literature sources (e.g., newspapers, websites) and non-medical databases (e.g., social science databases) to better understand the diverse clinical neuroscience KT activities used on social media.

One main factor that prevents the field of social media KT research from progressing further is that there is no standard or agreed upon metric to measure the success of a KT social media strategy [29]. A case study by Elliott et al. (2020) reported that there is an ongoing debate within this field regarding which KT metric(s) best indicate the success of a KT intervention. Additionally, while social media metrics (e.g., number of views) can provide information on the reach achieved by a social media KT strategy, improved social media reach does not necessarily indicate that the goals of KT such as improved knowledge or behaviour change are achieved [29]. Therefore, standard methods to measure the impact and success of social media KT strategies are needed to advance this field of research.

## Conclusion

This review found that YouTube, Facebook, Twitter, and a combination of multiple social media platforms are used to translate clinical neuroscience information to patients, HCPs, caregivers, and the general public. The clinical neuroscience information disseminated included content on dementia, MS, and concussion. While most studies found that disseminating clinical neuroscience information via Facebook, Twitter, and YouTube were effective at achieving the goals of KT in the target population, these results must be interpreted with caution due to the small number of included studies, low quality of evidence, and self-reported measures of KT. The field of the use of social media in the KT of clinical neuroscience is a nascent one and future work should attempt to improve the quality of research by addressing the significant sources of bias that we have identified.

## Supporting information

**S1 File. Search strategy and database search strings used to conduct the literature search on July 16, 2021.**
(DOCX)

**S2 File. PRISMA (Preferred Reporting Items for Systematic Reviews and Meta-Analyses) Checklist.**
(DOC)

**S1 Table. List of studies excluded at full-text screening with brief reasoning (N = 17).**
(PDF)

**S1 Fig. Traffic plot depicting the risk of bias assessments of non-randomized studies based on Cochrane's Risk of Bias in Non-randomized Studies – of Interventions (ROBINS-I) tool.**
(TIF)

**S2 Fig. Traffic plot depicting risk of bias assessments for one randomized controlled trial (parallel-group design) based on Cochrane's risk-of-bias tool for randomized trials (RoB 2).**
(TIF)

**S3 Fig. Summary plot of risk of bias assessments for non-randomized studies based on Cochrane's Risk of Bias in Non-randomized Studies – of Interventions (ROBINS-I) tool.**
(TIF)

**S4 Fig. Summary plot of risk of bias assessment for one randomized controlled trial (parallel-group design) based on Cochrane's risk-of-bias tool for randomized trials (RoB 2).**
(TIF)

**S5 Fig. Types of included knowledge translation proxy outcomes (by percentage).**
(TIF)

**S6 Fig. Types of social media platforms used for all included outcomes.**
(TIF)

**S7 Fig. Types of clinical neuroscience information disseminated for all included outcomes.**
(TIF)

## Author contributions

**Conceptualization:** Melissa Fazari, Michael David Cusimano.

**Data curation:** David Lightfoot.

**Formal analysis:** Rachel Victoria Baran, Melissa Fazari.

**Investigation:** Rachel Victoria Baran, Melissa Fazari.

**Methodology:** Rachel Victoria Baran, Melissa Fazari.

**Project administration:** Rachel Victoria Baran, Melissa Fazari.

**Resources:** Rachel Victoria Baran, Melissa Fazari.

**Software:** Rachel Victoria Baran, Melissa Fazari.

**Supervision:** Melissa Fazari, Michael David Cusimano.

**Validation:** Rachel Victoria Baran, Melissa Fazari.

**Visualization:** Rachel Victoria Baran, Melissa Fazari.

**Writing – original draft:** Rachel Victoria Baran, Melissa Fazari.

**Writing – review & editing:** Rachel Victoria Baran, Melissa Fazari, Michael David Cusimano.

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
