## [Decision Letter · Decision Letter 0]

24 Sep 2024

PDIG-D-24-00056

Social media strategies used to translate knowledge and disseminate clinical neuroscience research to healthcare users: A systematic review

PLOS Digital Health

Dear Dr. Cusimano,

Thank you for submitting your manuscript to PLOS Digital Health. After careful consideration, we feel that it has merit but does not fully meet PLOS Digital Health's publication criteria as it currently stands. Therefore, we invite you to submit a revised version of the manuscript that addresses the points raised during the review process.

Please submit your revised manuscript within 30 days Oct 24 2024 11:59PM. If you will need more time than this to complete your revisions, please reply to this message or contact the journal office at digitalhealth@plos.org. Please include the following items when submitting your revised manuscript:

We look forward to receiving your revised manuscript.

Kind regards,

PLOS Digital Health

Journal Requirements:

Additional Editor Comments:

Dear Dr. Cusimano, 

Thank you for choosing to submit your work with us. 

Your paper has been reviewed by 3 independent reviewers, in general the reviewers find the manuscript well written and easy to follow. The methodology proposed is transparent and rigorous, follows the accepted approach and guidelines for conducting a systematic review, starting with registering the protocol through PROSPERO, and reporting that follows the PRISMA checklist. 

However, some concern was pointed regarding including the study by Narayanaswami et al.(reference 23), the group in this study seem to be exposed to both social media platforms and traditional methods, whereas in your result suction you detail excluding studies where the intervention was disseminated via both social media and other methods (line 221), please clarify including this study. 

The major feedback is on your interpretation of the results and your discussion. Giving the small numbers of studies found and the low quality of those studies, drawing conclusions on the effectiveness of certain interventions should be stated with caution. Furthermore, a critical review of what is missing in the field, based on the analyzed papers, would be a valuable contribution. Specifically, it would be beneficial to discuss what is preventing the field from gaining momentum. The limited findings of the study should be used to highlight existing gaps and suggest actionable steps.

Given the nature of this study as a systematic review, we understand the limitations of conducting a quantitative assessment of the findings. We appreciate the narrative synthesis and summary of the results based on the existing knowledge. While some reviewers noted the absence of a quantitative synthesis as a limitation, the editors acknowledge that this is not feasible for this particular study and believe the narrative synthesis approach is appropriate. 

Concerns were raised about the scope of the databases searched and how comprehensive they are in answering the research question raised in this paper- clarify the rational of choosing the current databases, and possible limitations of expanding into other databases.

Below are the detailed comments provided by the reviewers for your consideration. 

Reviewers' comments:

Reviewer's Responses to Questions

**Comments to the Author**

1. Does this manuscript meet PLOS Digital Health’s publication criteria ? Is the manuscript technically sound, and do the data support the conclusions? The manuscript must describe methodologically and ethically rigorous research with conclusions that are appropriately drawn based on the data presented.

Reviewer #1: Yes

Reviewer #2: Partly

Reviewer #3: Yes

2. Has the statistical analysis been performed appropriately and rigorously?

Reviewer #1: N/A

Reviewer #2: No

Reviewer #3: N/A

3. Have the authors made all data underlying the findings in their manuscript fully available (please refer to the Data Availability Statement at the start of the manuscript PDF file)?

Reviewer #1: Yes

Reviewer #2: Yes

Reviewer #3: Yes

4. Is the manuscript presented in an intelligible fashion and written in standard English?

PLOS Digital Health does not copyedit accepted manuscripts, so the language in submitted articles must be clear, correct, and unambiguous. Any typographical or grammatical errors should be corrected at revision, so please note any specific errors here.

Reviewer #1: Yes

Reviewer #2: Yes

Reviewer #3: Yes

5. Review Comments to the Author

Please use the space provided to explain your answers to the questions above. You may also include additional comments for the author, including concerns about dual publication, research ethics, or publication ethics. (Please upload your review as an attachment if it exceeds 20,000 characters)

Reviewer #1: 

Thank you for the opportunity to peer review this paper. The manuscript aims to explore which social media strategies have been used to translate knowledge and disseminate clinical neuroscience research to healthcare users and how effective these strategies are. The scope was clear, and the authors successfully achieved this aim by providing a summary and evaluation of current social media strategies used to translate clinical neuroscience knowledge. The authors found that 80% of strategies assessed in studies found were effective based on several metrics and different social media strategies contributed to different outcomes. The limitations were acknowledged accordingly, future directions helpful and implications clear and strong. Overall, the paper is well-written and organized in a systematic manner. This manuscript contributes to the literature on a pertinent and timely topic. Congratulations on the well-done work.

Reviewer #2: 

Thank you for inviting me to review this paper.

I believe that disseminating scientific knowledge to a broader audience, particularly by identifying effective strategies for conveying this knowledge to patients and their families, represents a significant research problem. Therefore, I consider the general objective of this paper important. Specifically, exploring which strategies for transferring research findings in neuroscience (or any other scientific discipline) are effective is a crucial area of research.

However, to achieve these objectives, the current study would have to significantly improve the sampling strategy, increase the number of papers included, and define the research problem and intended contribution in more detail.

First, this study is ambitious in claiming its intended contribution to synthesize the overall evidence on the effectiveness of social media strategies in translating research knowledge. To show the effectiveness of different knowledge transfer strategies on social media, the study should have surveyed social media itself to show the comparison between different strategies in communicating research findings. Instead, this study focuses only on five papers published in databases like Web of Science and how those papers assess the effectiveness of social media dissemination. Therefore, I recommend that the focus of the paper and intended contributions be revised and defined more concretely. 

More concretely, process and keywords in S1 file reveals that the more appropriate research question should have been: how do selected neuroscience studies, published in medical databases, choose to disseminate their findings on social media? Still, there are too few studies selected to make any generalizations regarding different strategies these studies may have used, or to conclude whether social media strategies are more effective than other dissemination strategies or which social media channels are more effective than others.

Second, neuroscience study findings are often disseminated on social media through publishing houses, newspapers, or university websites. Those sources are not included in the analysis here. Instead, only the studies that analyzed dissemination through social media for raising education or awareness seem to be included (note the selection from almost 4000 papers to ultimately 5). Consequently, the research problem seems related to the quality of the published evidence on the impact of neurological research dissemination via social media to patients and other stakeholders. The evidence shows that the current quality of these five studies is weak, possessing significant potential biases and often lacking important methodological rigor (no control group, no control variables like prior social media savviness, etc.).

Third, for broader generalizations, more studies should have been selected and a broader set of databases may be relevant to use. For example, the selection of databases poses a problem because only selected medical databases were used (note that PubMed is not included among the databases, presumably because it overlaps with Web of Science, but the overlap is not concrete). Importantly, most of the evidence on social media dissemination of research findings would likely be published in social science databases related to business studies, education studies, etc.

Fourth, I recommend that the authors define more concretely the main research concepts, like knowledge transfer. What exactly is knowledge transfer and how would it be defined—with which metrics and in which concepts of interest? For example, some of the excluded studies in S1 Table could also be seen as studies of knowledge transfer. Other studies that deal with knowledge transfer in the medical domain could be an important baseline for developing a clearer definition of the research focus. For example, one of the recent influential studies on the effectiveness of disseminating medical research findings on social media has been published in Scientific Reports: Motta, M., Liu, Y., & Yarnell, A. “Influencing the influencers:” a field experimental approach to promoting effective mental health communication on TikTok. Sci Rep 14, 5864 (2024). https://doi.org/10.1038/s41598-024-56578-1.

Fifth, the selection of the dependent variable of interest is very important when evaluating the merits of the study. The change in knowledge levels or awareness is a very subjective measure that in these particular studies by default would have been inflated because this question is answered only by respondents who took part in the study and actively responded to the study questions. Being exposed to information, stimuli, or evidence would naturally lead to respondents being more aware (and knowledgeable). Instead, it is important to analyze the results of the studies using counterfactuals and comparisons with a control group or alternative channels to show the effectiveness of social media channels. It is interesting to note that the study that has done such a comparison more reliably has found no significant improvement of social media channels over other forms of dissemination. I would, therefore, caution authors to be careful about making overly positive conclusions based on these remaining four studies.

In conclusion, I would encourage first defining more concrete research questions, defining what the main concepts are and how they could be measured, and extending the scope of the research to include more studies either by expanding the medical domains or by changing the research question to include more studies that are currently excluded. Finally, I would recommend focusing on the quality of the evidence and providing a more critical review of why we do not know much about the effectiveness of disseminating knowledge about medical research on social media: publication bias, lack of outreach to a broader audience, low interest of researchers, etc. The science of science can provide useful ways of approaching these questions.

Reviewer #3: 

Title: Social media strategies used to translate knowledge and disseminate clinical neuroscience research to healthcare users: a systematic review

1. Summary of the Manuscript:

The paper systematically reviews the effectiveness of social media strategies in translating clinical neuroscience knowledge to healthcare users, which include patients, healthcare providers, caregivers, and the public. The review considers six electronic databases up to January 29, 2024, and includes five peer-reviewed studies that examine platforms like YouTube, Facebook, and Twitter. The findings suggest that most strategies involving interactive, recent, and consistent content dissemination are effective in knowledge translation, enhancing uptake, awareness, understanding, and behavior change.

2. General Comments:

The paper addresses a timely topic by exploring the effectiveness of social media strategies in translating clinical neuroscience knowledge, specially given the increasing reliance on digital platforms for health information. The inclusion of studies up to January 2024 provides a current perspective on the topic.

The paper is well-organized, with a clear structure that includes a comprehensive background, detailed methodology, and well-explained results and discussion sections. The authors acknowledge the limitations of the included studies, such as small sample sizes and varying evaluation metrics. This transparency is appreciated and highlights the need for more standardized and larger-scale studies in this area. However, these very reasons make me recommend the authors to slightly change the perspective they are tackling the issue in the current version of the paper.

3. Specific comments

a. Title & abstract

The title reflects well the content and aim of the paper. The abstract provides a good overview. In general the rest of the paper is well-written and easy to follow.

Line 68: There is lack of evidence from what is explained in the abstract for this statement, which suggests a general rule from very few studies.

b. Introduction:

The introduction is clear and guides the reader nicely through the main topic of the paper. The goals of the paper are stated concisely and clearly.

c. Methods:

The eligibility criteria are sound, and follow a well-known PICO methodology. The fact that two independent reviewers reached a Cohen’s kappa of 1.0 seems to indicate the task was very simple. Please provide further information about the way annotators found a consensus regarding the overall ROB judgements. This is arguably one of the most important things to discuss in the paper given the potential implications of overlooking biases present in our data sources. 

How is the summarization process carried out? Manually? Using any automatic system?

The review of a given study, is performed according to the summary, or the full-text? In case it is the former, please describe in detail who or how it was created. The inclusion in the methodology of these summaries is inevitably introducing another layer of complexity. 

d. Results:

Lines 218-223 could benefit from some rephrasing, as they are currently difficult to understand. 

I appreciated the “Risk of Bias Assessment” section; it is well explained and written. However, I am not sure that Figures 2, 3, and 4 add much value. Is there another way to present these ratios?

The results for each outcome are presented in an orderly manner, but the authors report on them in a rather shallow way, which prevents them from drawing any meaningful and novel conclusions. I recommend that the authors consider aspects of the reviewed papers that may be compared and analyzed collectively to improve knowledge on the topic. 

Please include the number of patients in each study or outcome in the presentation of results. Additionally, information about the demographics would be greatly appreciated, if available.

e. Discussion:

The authors acknowledge the small size of the sample studied, as well as the difficulties arising from the fact that each study employs a different set of metrics that are not directly comparable. In some cases, these metrics are self-reported assessments, making it difficult to evaluate the quality or impact of different KT strategies when comparing studies, as done here. This is problematic when the authors conclude that some strategies are successful based on 4 out of 5 studies, given that the metrics used in these studies are not equally valuable. More work is needed to disentangle the information each evaluation method brings to the discussion in the context of its study.

The analysis of the possible reasons why the strategies used in one of the studies did not work is robust, sound, and easy to follow. However, I wonder whether the fact that the materials in this case were released prior to dissemination via social media makes it unsuitable for this review. Other studies were dismissed because KT was evaluated after exposure to both social media and other channels, which is also the case here. I would like to understand better why this study was included while others were rejected. The authors state, “Future studies should control for important confounding factors such as these to minimize risk of bias.” This perspective could make for a great version of this paper. A deeper analysis of what is currently missing in the analyzed papers, and what prevents the field from gaining momentum, would contribute significantly, not only in this context but in other domains as well.

I congratulate the authors on the rest of the discussion.

f. Conclusion:

The conclusions of this study do not introduce any novel ideas. The support for claiming that certain social media platforms are useful conveyors of knowledge translation (KT) is based on too few studies with evident insufficiencies, such as small sample sizes and inadequate analysis of the demographics and representativeness of the target populations.

The authors claim that three specific social media platforms have been found effective, but this conclusion is drawn from only 4 out of 5 studies. Moreover, these platforms have only been found effective compared to the absence of social media use, rather than in comparison to other social media platforms. This leaves a significant gap, making the inference of broader conclusions problematic.

While the authors acknowledge their results as preliminary and encourage further research, one of the conclusions seems to contradict one of the main results. They state, “our review suggests that the method by which content is created and disseminated via social media is of greater importance than the specific target audience and social media platforms used to disseminate clinical neuroscience information.” This appears to contradict recommendation (b) in the “Implications” section.

g. References:

In reference numbers 27 & 28: please indicate the date you accessed the site for the last time.

4. Strengths of the Manuscript:

The paper is easy to follow, provides a good introduction to the topics involved, and outlines the results obtained in 5 studies of different nature, all of them about the leverage of digital strategies in social media to improve KT.

The paper is structured in an orderly manner, and describes the methodology well, which is robust and strict.

5. Weaknesses of the Manuscript:

The main problem with this review is that too few studies were considered. Additionally, the studies do not share a common evaluation metric; instead, each employs a different set of outcomes and types of assessment. This complicates the inference of any general conclusion without a better understanding of the population involved in each study and what they have in common with the other studies considered. Furthermore, all the studies seem to be tied to predominantly Western social media, thus limiting the scope of their results.

In its current form, the paper is too limited to a description of the main results as reported by the authors of the original papers, hence lacking originality in its conclusions. 

6. Recommendations for Improvement:

In the end, I mostly noticed some apparent contradictions between the conclusions and some of the main results outlined in the discussion section, probably due to a lack of depth in the analysis (see previous comments). There is an insufficient analysis of aspects related to the population, the specific content of the campaigns, the type of social media… considered in each study that would provide an interesting perspective and would add substantial value to this paper.

I suggest the authors tackle the paper from a slightly different perspective, pointing out current issues that may be hindering the ability to further progress in this field. Authors have done a substantial amount of work that might be leveraged by other researchers to agree on standards (population sizes, demographics to be used in studies, type of content to be investigated, evaluation metrics…).

7. Recommendation:

The paper in its current form does not provide novel insights but rather presents some characteristics of the reviewed papers, which are few. Several aspects of the studies considered have not been discussed, which would shed light on why some campaigns appeared to work better than others. These aspects are largely overlooked in the current submission.

6. PLOS authors have the option to publish the peer review history of their article (what does this mean? ). If published, this will include your full peer review and any attached files.

**Do you want your identity to be public for this peer review?** For information about this choice, including consent withdrawal, please see our Privacy Policy .

Reviewer #1: No

Reviewer #2: No

Reviewer #3: No

---

## [Editor Report · Decision Letter 1]

10 Feb 2025

Social media strategies used to translate knowledge and disseminate clinical neuroscience research to healthcare users: A systematic review

PDIG-D-24-00056R1

Dear Dr. Cusimano,

We are pleased to inform you that your manuscript 'Social media strategies used to translate knowledge and disseminate clinical neuroscience research to healthcare users: A systematic review' has been provisionally accepted for publication in PLOS Digital Health.

Best regards,

PLOS Digital Health